# Characterization of a Robust and pH-Stable Tannase from Mangrove-Derived Yeast *Rhodosporidium diobovatum* Q95

**DOI:** 10.3390/md18110546

**Published:** 2020-10-30

**Authors:** Jie Pan, Ni-Na Wang, Xue-Jing Yin, Xiao-Ling Liang, Zhi-Peng Wang

**Affiliations:** 1Qingdao Mental Health Center, Qingdao University, Qingdao 266034, China; zxy18615349853@163.com (J.P.); WFB18653266241@163.com (N.-N.W.); hbc2369031698@163.com (X.-J.Y.); 2Marine Science and Engineering College, Qingdao Agricultural University, Qingdao 266109, China; wangzp@qau.edu.cn

**Keywords:** tannase, pH stability, tannin biodegradation, gallic acid

## Abstract

Tannase plays a crucial role in many fields, such as the pharmaceutical industry, beverage processing, and brewing. Although many tannases derived from bacteria and fungi have been thoroughly studied, those with good pH stabilities are still less reported. In this work, a mangrove-derived yeast strain *Rhodosporidium diobovatum* Q95, capable of efficiently degrading tannin, was screened to induce tannase, which exhibited an activity of up to 26.4 U/mL after 48 h cultivation in the presence of 15 g/L tannic acid. The tannase coding gene *TANRD* was cloned and expressed in *Yarrowia lipolytica*. The activity of recombinant tannase (named TanRd) was as high as 27.3 U/mL. TanRd was purified by chromatography and analysed by SDS-PAGE, showing a molecular weight of 75.1 kDa. The specific activity of TanRd towards tannic acid was 676.4 U/mg. Its highest activity was obtained at 40 °C, with more than 70% of the activity observed at 25–60 °C. Furthermore, it possessed at least 60% of the activity in a broad pH range of 2.5–6.5. Notably, TanRd was excellently stable at a pH range from 3.0 to 8.0; over 65% of its maximum activity remained after incubation. Besides, the broad substrate specificity of TanRd to esters of gallic acid has attracted wide attention. In view of the above, tannase resources were developed from mangrove-derived yeasts for the first time in this study. This tannase can become a promising material in tannin biodegradation and gallic acid production.

## 1. Introduction

Tannins are a class of poly-phenolic compounds widely existing in higher plants [1]. Among all the plant constituents, they rank the fourth in abundance, next to cellulose, hemicellulose, and lignin [2]. Benefiting from their forming of strong complexes with proteins and many other biological macromolecules, their bad taste, as well as their indigestibility, tannins provide defence against invading microbial pathogens and herbivores [2]. However, there still exist some microorganisms that are resistant to them [3]. These microorganisms were able to degrade tannins with the help of secreted enzymes. During the process, tannins were transformed into molecules that are available energy or carbon sources for cell growth and metabolism [3].

Tannase (i.e., tannin acyl hydrolase, EC 3.1.1.20) is a kind of hydrolase targeting the ester bonds in ellagitannins, gallotannins, complex tannins, and gallic acid esters, releasing gallic acid, ellagic acid, etc. [3,4,5,6]. It has become increasingly important in the pharmaceutical industry to produce gallic acid from gallotannins [7,8,9,10,11,12,13]. As an essential drug intermediate, gallic acid has mainly been employed in the synthesis of trimethoprim (an antibacterial agent) and gallate esters (preservatives in the food production industry) [10,12]. In addition, dissociative gallic acid is also attractive in terms of its biological properties, especially for its antibacterial, antioxidant, antiallergic, anticarcinogenic, and anti-inflammatory activities [10,11]. The enzymatic method, a green alternative, is often utilized to convert gallotannins into gallic acid, which not only overcomes the drawbacks of traditional acid hydrolysis but also increases the product purity and yield [11,12,13,14].

In previous studies, many tannases derived from bacteria and fungi have been well characterized [3,4,5,6]. Considering the good thermal stability and the enzyme yield, tannases were mainly produced by fungal fermentation in industry, especially by fermentation of the *Aspergillus* species [15,16,17,18]. High catalytic activities of these tannases were maintained in a narrow pH range (about 5.0–7.0) [15,16,17,18,19,20]. However, for the sake of better action on acidic substrates and collection of hydrolysis product, developing tannases stable at low pH is necessary, which can facilitate gallic acid production [7,8,9,10,11,12,13]. Although the comparison of tannases from bacteria and fungi has been conducted, tannases that are efficient and stable at low pH are still rare [3,4,5,6]. Previous studies demonstrate that tannases from yeast species have been only verified in *Sporidiobolus ruineniae*, *Candida* sp., *Aureobasidium*, *Blastobotrys adeninivorans*, and *Kluyveromyces marxianus* [11,21,22,23,24]. Thus, uncharacterized tannases from yeast are proven potential resources that need to be developed.

As an ecotone ecosystem, mangroves are enormously abundant in microbial species, including yeasts [25,26]. Alongi et al. found that yeasts were able to synthesize enzymes functioning in the degradation of plant materials [25,26]. In this study, we screened a tannin-degrading yeast strain *Rhodosporidium diobovatum* Q95. Subsequently, the gene encoding the obtained tannase (named TanRd) was cloned, followed by its expression in a food-grade host, *Yarrowia lipolytica* (Fungi). The recombinant tannase featured excellent pH stability and distinct robustness, which made it a potent tool in tannin biodegradation and gallic acid production.

## 2. Results

### 2.1. Strain Q95 Derived from Mangrove Has the Ability to Degrade Tannic Acid at Low pH

As a variety of mangrove plants are rich in tannins, mangrove-derived microorganisms are expected to induce tannase with good properties [26]. Sixty-four yeast strains isolated from mangrove samples were first purified and then inoculated on YPT (Yeast extract-Peptone-Tannic acid) plates at a pH of 4.5. Upon the transformation of tannic acid in the plates into gallic acid, the corresponding part showed a different color from that containing the original tannic acid. The size of the circle around the strain can act as a relative index for the evaluation of the degradation ability toward tannic acid [3,4,5,6]. As shown in Figure 1a, strain Q95 led to the biggest circle on the plate, indicating that the tannase secreted from it had the highest activity. As for the streak cultivation on the YPT plate (pH = 3.0), strain Q95 corresponded to the part with more obvious color change (Figure 1b,c).

To determine the tannase activity, the strain Q95 was cultivated in YPT mediums with different concentrations of tannin (5, 10, 15, 20, and 25 g/L). Figure 2a displays that the biomass showed an increasing trend with the tannin concentration. With the increase in tannin concentration, the tannase activity as revealed by accumulation of gallic acid in the medium, increased with tannin availability up to 15 g/L tannins. When the tannin concentration was 15 g/L, the detected tannase activity reached the maximum (26.4 U/mL), even higher than the activities of those originated from *Aspergillus* species [15,16,17,18]. HPLC analysis of the reaction mixture further verified that tannic acid can be converted into gallic acid in the catalytic system of tannase generated by strain Q95 (Figure 2b). To sum up, the yeast strain Q95 was an outstanding producer of tannase, efficient even at low pH, and tannase resources were developed from mangrove-derived yeasts for the first time in this study.

### 2.2. Identification of Strain Q95

The colony of strain Q95 on the YPD plate was light red with smooth edges (Figure 1c). Observed under a microscope, the strain was found to have the characteristics of yeast. Besides, the cells were oval, with individual ones budding at one end (data not shown). Further, the ITS (Internal transcribed space)) rDNA of strain Q95 was sequenced to identify the strain, after which the sequence comparison was conducted with BLAST (Basic local alignment search tool). It was found that the ITS rDNA sequence (Accssion: MW173231) had the highest similarity to that of the identified strain *Rhodosporidium diobovatum* Q95. However, clustering analysis gave the information that the strain Q95 was on the same branch as *Rhodosporidium* (Figure 3). The BLAST results showed that the ITS rDNA sequence of strain Q95 had 97.8% identity to that of type strain *Rhodosporidium diobovatum* CBS 6085 (KY104768.1). Jointly verified by the BLAST and phylogenetic tree results, the strain Q95 was from the species *Rhodosporidium diobovatum*. 

### 2.3. Bioinformatics Analysis of TanRd 

To clone the gene encoding TanRd, we sequenced the genome DNA of Q95. The sequence analysis showed that there existed a putative gene encoding TanRd, which was designated as *TANRD* (Submission ID: 2395723). The open reading frame (ORF) consisted of 1719 bp and encoded a protein composed of 572 amino acids. Further bioinformatics analysis was also performed. The results indicated that the first 19 amino acids of TanRd were predicted as the signal peptide, which was in conformity with the secretion characteristic. The signal peptide had a theoretical isoelectric point (pI) of 5.24 and the mature protein had a molecular weight (Mw) of 59.7 kDa. According to BLAST alignment on NCBI (National Center for Biotechnology Information), one conserved domain was found in TanRd, which belonged to the tannase and feruloyl esterase family [5,15,16,17,18]. To further explore the attribution of TanRd, a phylogenetic tree was constructed based on the amino acid sequences of it and other reported tannases. Figure 4 shows that the TanRd was a kind of fungal tannase, distinctly different from bacterial tannases [5]. TanRd was mostly close to TanA, a characterized tannase from *Aspergillus oryzae* [8]. In addition, TanRd was found to have the typical conserved “CS-D-HC motif” existing in the biochemically distinct members of the tannase family [15,16,17,18,27,28,29].

### 2.4. Expression of TanRd in Y. lipolytica

In the present study, TanRd was expressed in *Y. lipolytica*, a common heterologous host realizing remarkable extracellular secretion [30]. The activity of recombinant tannase was as high as 27.3 U/mL. The purified TanRd protein from the supernatant was subjected to SDS-PAGE (sodium dodecyl sulfate–polyacrylamide gel electrophoresis) analysis. Figure 5a demonstrates a distinct single band appearing in the lane, from which the Mw of TanRd was determined to be around 75.1 kDa. However, the secreted protein with His-tag had the theoretical Mw of 59.7 kDa. This discrepancy probably resulted from the glycosylation at the amino acid region. Several N-glycosylation recognition sites of TanRd were predicted using NetNGlyc 1.0 server. The glycosylation, common to fungal tannases, may be related to the good thermal stability of tannases [15,16,17,18,31]. However, it has been reported that the N-glycosylation has no effect on the activity and stability of AoTanB from *A. oryzae* [31]. The specific activity of TanRd toward tannic acid was detected as 676.4 U/mg.

### 2.5. Temperature Properties of TanRd

The enzymatic properties of the purified TanRd at different temperatures were examined. As shown in Figure 5b, its activity peaked at 40 °C and was maintained higher than 70% at 25–60 °C. As for the thermostability, the enzyme activity without 12-h incubation was defined as 100%. TanRd was rather stable at below 50 °C (Figure 5c). Specifically, about 80% of the activity remained after 12 h incubation at 50 °C. However, its activity declined greatly when the temperature exceeded 60 °C and even completely ceased at above 65 °C. As reported in the literature, microbial tannases generally have their optimal temperature in the range of 20–60 °C and are thermally stable at 30–60 °C. Notably, fungal tannases exhibit higher activity and stability than bacterial and yeast tannases under a variety of temperature conditions [15,16,17,18]. Usually limited by the thermal stability, the industrial applications of enzymes may make achievements when adopting the more thermostable tannase. TanRd can be a thermostable alternative choice.

### 2.6. pH Properties of TanRd

The activity of TanRd at different pH was also investigated. As shown in Figure 6a, over 60% of themaximal activity was maintained in a broad pH range (2.5–6.5). As for the pH stability, the enzyme activity without 12-h incubation was defined as 100%. Besides, TanRd retained over 65% of its maximal activity following 12 incubation at a pH range from 3.0 to 8.0. Surprisingly, above 40% of the activity still remained after the incubation in the broader pH range (2.0–9.5) in this research (Figure 6b). The optimal pH values of most fungal tannases are located around 6.0, whereas those of bacterial tannases mostly lie between 7.0 and 9.0. The enzyme derived from *K. marxianus* had an optimal pH of 4.5 in vitro and high stablity at pH 4.0–4.5, while the optimal pH of tannase from *Candida* sp. was found at 6.0 [11,22]. The cell-associated tannase from *S. ruineniae* was stable, reflected by the retainment of more than 80% of the activity at pH from 5.0 to 9.0 [32]. Since the substrate and product of tannase are both acidic, the TanRd screened in this study can facilitate gallic acid production owing to its unique pH-stable property. Compared with other reported fungal tannases, TanRd had much better pH properties (Table 1). Therefore, the tannase characterized in this study can function as a potent tool in tannin biodegradation and gallic acid production.

### 2.7. Effects of Ions on the TanRd Activity 

As indicated by Figure 7, metal ions at 1 mM did not exert a great inhibitory effect on the TanRd activity, except for Cu^2+^ and Ba^2+^ (Figure 7). The relative activity of TanRd was 141.5% and 136.2% in the presence of Mn^2+^ and Co^2+^, respectively. However, in the case of ME (2-hydroxy-1-ethanethiol) at 1 mM, the relative activity was remarkably reduced to 16.3%. This significant inhibition caused by ME has also been found in several reported tannases, the reason for which is the capability of ME to destroy the disulfide bonds in tannase structure. When the concentration was increased to 10 mM, the situation became different. The activity of TanRd was inhibited by Cu^2+^, Al^3+^, Ba^2+^, Ca^2+^, and SDS. It is worth noting that Mn^2+^ and Co^2+^ strongly promoted the activity. In short, TanRd possessed great metal ion tolerance and Mn^2+^ and Co^2+^ can be adopted as good activating agents for TanRd. 

### 2.8. Substrate Specificity of TanRd 

The purified TanRd was incubated with several esters of gallic acids, respectively, to investigate its substrate specificity, with the results shown in Table 2. To be specific, TanRd exhibited lower specific activity toward natural substrates (ECG and CG) than toward synthetic substrates (PG). The superiority of TanRd to other tannases in terms of ECG and CG degradation capacities suggests that it can be also used for tea treatment or the extraction of bioactive substance from tea leaves [17,35]. 

## 3. Materials and Methods

### 3.1. Materials, Strains, and Mediums 

The tannase-producing strains were cultured in YPT medium, which was prepared with seawater and contained 5 g/L peptone, 10 g/L glucose, 10 g/L tannic acid, 5 g/L (NH_4_)_2_SO_4_, and 0.001 g/L bromophenol blue, pH = 3.0 [3,4,5,6]. The uracil mutant *Y. lipolytica* URA-strain and expression vector pINA1312 were both kindly provided by Zhenming Chi, Ocean University of China. *Y. lipolytica* URA-transformants were screened on a YNB (yeast nitrogen base) plate, which consisted of 1.7 g/L yeast nitrogen base without amino acids, 5.0 g/L (NH_4_)_2_SO_4_, 10.0 g/L glucose, and 25.0 g/L agar [30]. To prepare GPPB medium, 30.0 g/L glucose, 2.0 g/L yeast extract, 1.0 g/L (NH_4_)_2_SO_4_, 2.0 g/L KH_2_PO_4_, 0.1 g/L MgSO_4_·7H_2_O, and 3.0 g/L K_2_HPO_4_ was added, with a pH of 6.8 [30].

### 3.2. Screening Tannase-Producing Strains at Low Temperature

Mangrove plant samples were first milled and then spread on YPT plates [26]. Subsequently, the plates underwent a 48 h incubation at 25 °C. Next, a total of 64 single colonies were selected and transferred to the new YPT plates. After that, the colony of strain Q95was transferred into YPT liquid medium and incubated at 25 °C for 48 h. The obtained culture was then subjected to centrifugation at 5000× *g* in order to determine the activity of tannase in the supernatant. The TanRd activity was evaluated depending on the gallic acid released at 40 °C with 0.5% propyl gallate as the substrate. A spectrophotometric method based on the chromogen formation between gallic acid and rhodanine was adopted to estimate the amount of gallic acid [27]. The absorbance was detected against the inactivated enzyme as blank at 520 nm. One unit of TanRd activity was defined as the amount required to produce 1 μmol gallic acid per minute of reaction. 

### 3.3. Strain Identification

ITS rDNA of strain Q95 underwent PCR amplification with universal primers ITS5 5′-TCCGTAGGTGAACCTGCGG-3′) and ITS3 (5′-TCCTCCGCTTATTGATATGC-3′), followed by its sequencing and the comparison with other ITS rDNA sequences by BLAST (National Center for Biotechnology Information, Bethesda, USA). The phylogenetic tree was created with the neighbor-joining method on the basis of closely related sequences using MEGA version 7.0 (Center for Evolutionary Medicine and Informatics, The Biodesign Institute, Tempe, AZ, USA). 

### 3.4. Bioinformatics Analysis of TanRd 

With the aim of identifying the gene encoding TanRd, the sequencing and annotation of the genomic DNA of strain 95 were performed by Novogene, China. HMMER3 was applied to the comparison of the gene protein sequence with the CAZy (Carbohydrate-Active enZYmes) database, from which the annotation information of carbohydrate-active enzyme annotation was obtained [5]. The filter condition was set as E-value < 1 × 10^−5^. The sequence analysis indicated that there existed a putative gene encoding TanRd, with an ORF of 1719 bp. N-glycosylation sites are predicted using the NetNGlyc 1.0 server (http://www.cbs.dtu.dk/services/NetNGlyc/) (Technical University of Denmark, Lyngby, Denmark).

The signal peptide analysis was conducted with the SignalP 4.1 server (http://www.cbs.dtu.dk/services/SignalP-4.1/) (Technical University of Denmark, Lyngby, Denmark), and domain analysis was made in the Conserved Domain Database (https://www.ncbi.nlm.nih.gov/cdd) (National Center for Biotechnology Information, Bethesda, USA). Both the theoretical pI and Mw were predicted online (http://web.expasy.org/compute_pi/) (SIB Swiss Institute of Bioinformatics, Geneva, Switzerland). With reported tannases, the phylogenetic tree was generated by adopting the neighbor-joining method in MEGA version 7.0. 

### 3.5. Secretory Expression and Purification of TanRd 

After codons were optimized, the *TANRD* gene containing the XPR2 signal peptide gene was synthesized by Synbio Technologies, China. The average GC content of the optimized sequence was 50% codon and its adaptable index (CAI) was 0.84. Besides, there were no medium or large hairpins in it. The DNA fragment synthesized in this study was transformed into the URA-strain [30]. After the cultivation in GPPB liquid medium at 30 °C for 48 h, the positive transformants were detected for their tannase activities. Specifically, the recombinant strain T73 had the highest extracellular activity. During the T73 fermentation at flask, the TanRd activity and biomass were determined every 12 h. All the data were collected in triplicate [30]. The supernatant of strain T71 was adjusted until pH 7.5 and then loaded on a gel filtration chromatography column as well as an anion exchange chromatography column (GE Healthcare, Chicago, IL, USA). The TanRd can be attached to the gel and was then washed off. The Mw and purity of TanRd were verified based on SDS-PAGE on 12% (*w*/*v*) gel.

### 3.6. Effects of Temperature and pH on TanRd Activity and Stability 

The hydrolysis mediated by TanRd proceeded in 10 mM glycine–NaOH buffer (pH 4.5) at temperatures ranging from 20 °C to 70 °C, the results of which showed the optimal reaction temperature. To investigate the thermal stability of TanRd, the enzyme after purification was first incubated at temperatures ranging from 20 °C to 70 °C for 12 h and then the remaining activity at 40 °C was detected. Tannic acid solutions were prepared with 10 mM buffer at different pH levels (Na_2_HPO_4_–citric acid, pH 2.0–8.0; glycine–NaOH, pH 8.5–11.0) to act as the substrate of TanRd for the determination of the optimal reaction pH. pH stability was estimated according to the remaining activity after the incubation at 40 °C for 12 h in buffer at different pH levels. All reactions were allowed to proceed in triplicate.

### 3.7. Effects of Some Chemical Compounds, Metal Ions, and NaCl on TanRd Activity 

The stock solutions of the chemical compounds and metal ions were prepared first and then added to the tannic acid solutions to obtain the final concentrations of chemical compounds/metal ions of 1 mM and 10 mM, respectively. The TanRd-catalyzing reactions proceeded at 35 °C to investigate the effects of these substances on its activity. Meanwhile, these reactions were also performed in tannic acid solutions with different concentrations of NaCl (0–3.0 M) at 35 °C. The reaction taking place in the original tannic acid solution without other substances was used as a control. All reactions were allowed to proceed in triplicate. 

### 3.8. Determination of Gallic Acid and Ester by HPLC

The content of gallic acid and esters of gallic acids was determined at 278 nm by using HPLC (Agilent Technologies, CA, USA). The HPLC conditions were as follows: symmetry C18 column (3.0 mm × 250 mm, 5 m); mobile phase A (0.5% acetic acid aqueous solution) and mobile phase B (acetonitrile); mobile phase flow rate of 0.5 mL min^−1^; 45 min elution in a gradient manner, as previously described [28]. 

## 4. Conclusions

In this study, a novel robust and pH-stable tannase was screened, expressed extracellularly, and characterized. Its molecular weight was approximately 75.1 kDa and the specific activity reached 676.4 U/mg toward tannic acid. The highest activity was detected at 40 °C, and more than 70% of the activity was maintained at temperatures ranging from 25 °C to 60 °C. As for the pH stability, we found that it possessed over 60% of the activity in a broad pH range of 2.5–6.5. Moreover, the activity still exceeds 70% of the maximum after the incubation at pH 3.0–8.0. Additionally, TanRd had strong substrate specificity towards different esters of gallic acid. Therefore, the tannase characterized in this study can function as a potent tool in tannin biodegradation and gallic acid production.

## Figures and Tables

**Figure 1 marinedrugs-18-00546-f001:**
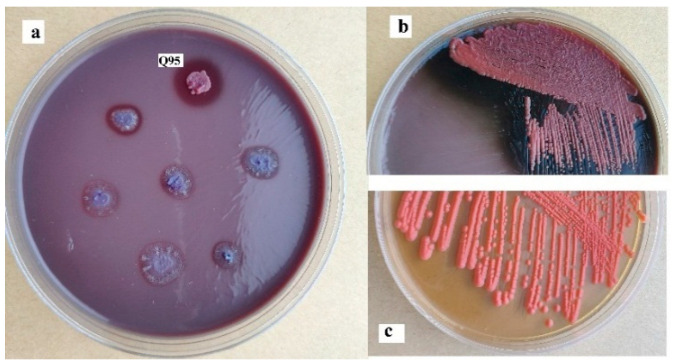
(**a**) Tannic acid-degrading strains isolated from mangrove samples on the YPT plate. (**b**) Strain Q95 incubated on the YPT plate by streak cultivation. (**c**) Strain Q95 incubated on the YPD (Yeast extract-Peptone-D-glucose) plate by streak cultivation as a control.

**Figure 2 marinedrugs-18-00546-f002:**
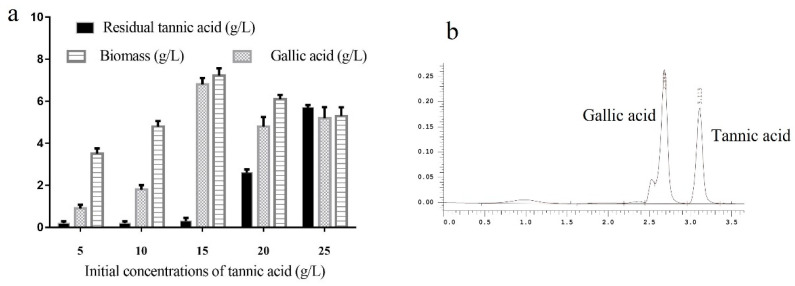
(**a**) Tannic acid degradation analysis of Q95 under the conditions of different tannin acid concentrations. (**b**) HPLC analysis of the substrate solution after tannase hydrolysis.

**Figure 3 marinedrugs-18-00546-f003:**
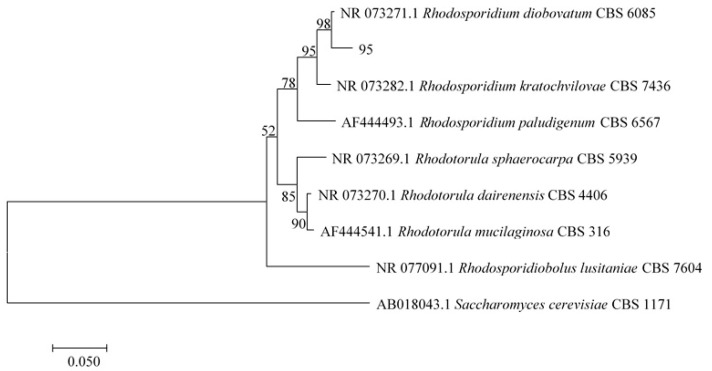
The phylogenetic tree generated with the neighbor-joining method based on the ITS rDNA gene sequences. Branch-related numbers are bootstrap values (confidence limits).

**Figure 4 marinedrugs-18-00546-f004:**
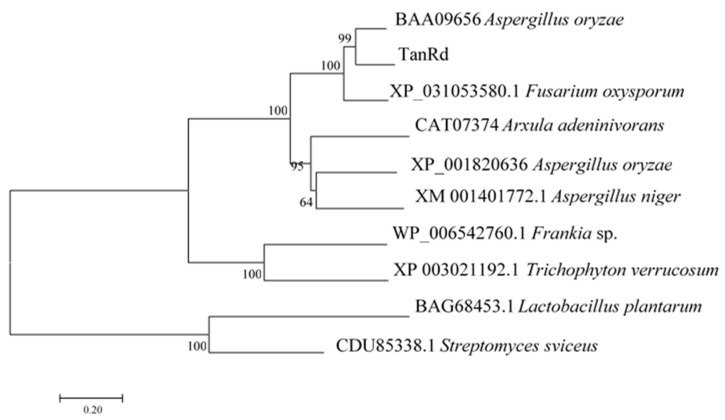
The phylogenetic tree generated with the neighbor-joining method based on the tannase amino acid sequences. Branch-related numbers are bootstrap values (confidence limits).

**Figure 5 marinedrugs-18-00546-f005:**
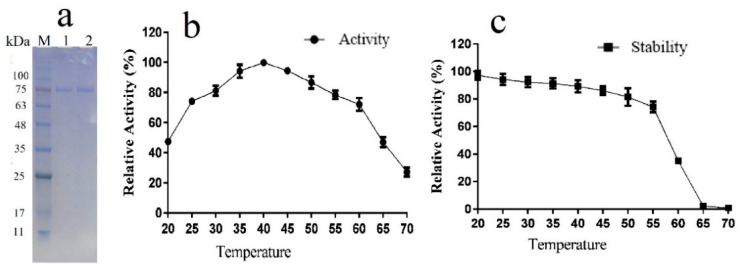
(**a**) Analysis of TanRd by SDS-PAGE. Lane M, pre-stained protein ladder; Lane 1 and lane 2purified TanRd. (**b**) Effect of temperature (°C) on the TanRd activity. (**c**) Effect of temperature (°C) on the TanRd stability. Data are expressed as mean ± standard deviation, *n* = 3.

**Figure 6 marinedrugs-18-00546-f006:**
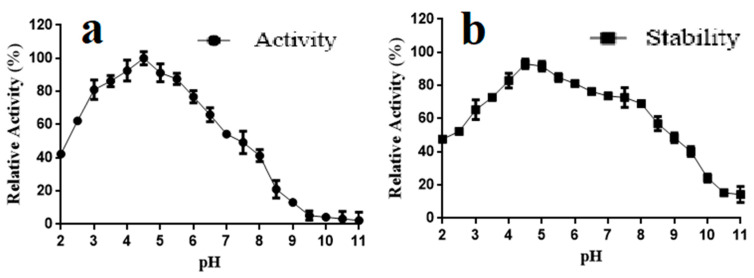
(**a**) Effect of pH on the TanRd activity. (**b**) Effect of pH on the TanRd stability. Data are expressed as mean ± standard deviation, *n* = 3.

**Figure 7 marinedrugs-18-00546-f007:**
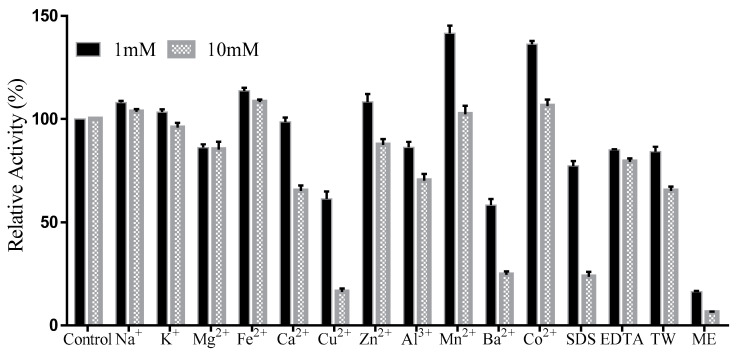
Effects of metal ions and chemicals on the TanRd activity. TW: tween 80; ME: 2-hydroxy-1-ethanethiol. Data are shown as mean ± standard deviation, *n* = 3.

**Table 1 marinedrugs-18-00546-t001:** Comparison of TanRd with reported tannases in terms of their properties. At optimal pH/temperature, TanRd showed the maximal activity. At pH-stable range, more than 60% activity remained after incubation.

Microorganisms	Optimal pH/Temperature (°C)	pH-Stable Range	Ref.
*Penicillium notatum*	5.0/35–40	3.0–8.0	[33]
*Emericella nidulans*	5.0/45	4.0–5.0	[34]
*Aspergillus phoenicis*	6.0/60	2.5–7.0	[34]
*Aspergillus niger*	6.0/80	3.0–8.0	[17]
*Sporidiobolus ruineniae*	7.0/40	5.0–9.0	[32]
*Kluyveromyces marxianus*	4.5, 8.5/35	4.0–6.0	[11]
*Rhodosporidium diobovatum*	4.5/40	3.0–8.0	This study

**Table 2 marinedrugs-18-00546-t002:** Comparison of the TanRd specific activity towards different esters of gallic acids.

Substrate	Specific Activity (U/mg)	Km (mM)
TA	676.4	1.87
PG	872.3	1.49
EGCG	572.1	2.19
ECG	721.3	1.65
CG	723.8	1.67
MG	473.6	2.31

PG: propyl gallate; EGCG: epigallocatechin gallate; TA: tannic acid; ECG: epicatechin gallate; CG: catechin gallate; MG: methyl gallate.

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
