# Peer review of "Characterization of a Robust and pH-Stable Tannase from Mangrove-Derived Yeast Rhodosporidium diobovatum Q95"

_marinedrugs, 2020, doi:10.3390/md18110546_

Round 1

Reviewer 1 Report

Pan et al. describe catalytic properties of a newly purified tannnase from a mangrove-derived yeast. This is a very straightforward study with basically accurate experiments and data, providing a slightly new finding. I consider this MS deserves publication in somewhere. My only concern is whether this MS is worth to be published in Marine drugs, considering its journal rank/impact factor in the research field. I cannot evaluate it because my research field is far from it.

Specific comments:

Around Line 70: Author can describe how many isolates have been tested/screened here, although it is described in Materials and Methods.

Fig. 2: B and C can be merged/overlapped so that readers can understand the results more easily.

Fig. 5: Positions of panels (b) and (c) may be strange. They may be put upper or upper-left from the figures, just like the case of Fig. 2. And, Figure legends for B and C are missing.

Data regarding the enzyme stability in Figs. 5C and 6B: Please show data about “relative enzyme activity” compared without incubation for 12 h before the reaction. In other words, please compare enzyme activity with/without 12-h incubation.   

Table 1: “Optimal pH/temparature” and “pH-stable range” should be defined (criteria for them).

Lines 269-270: Did the authors just add (but not adjust) each salt into GPPB medium? Because GPPB medium must contain some amount of salts, I suppose this should be described like “xx mM NaCl was “added” Or, did the authors measure/confirm the final concentration of each salt? Please clarify this.

Author Response

Pan et al. describe catalytic properties of a newly purified tannnase from a mangrove-derived yeast. This is a very straightforward study with basically accurate experiments and data, providing a slightly new finding. I consider this MS deserves publication in somewhere. My only concern is whether this MS is worth to be published in Marine drugs, considering its journal rank/impact factor in the research field. I cannot evaluate it because my research field is far from it.

Authors’ response:

We would like to thank you for the kindly help to revise this manuscript. We appreciated all the comments and suggestions.  The research in the MS exactly matches the Special Issue, "Marine Enzymes: Sources, Biochemistry and Bioprocesses for Marine Biotechnology – II".

Specific comments:

Around Line 70: Author can describe how many isolates have been tested/screened here, although it is described in Materials and Methods.

Authors’ response:

Thanks for your kind recommendation. The amount of the isolates was added.

Fig. 2: B and C can be merged/overlapped so that readers can understand the results more easily.

Authors’ response:

Thanks for your kind recommendation. The amount of the isolates was added. Because coVID-19 emerged in Qingdao and controls on the movement of people has been strengthened, we cannot get the original data in the HPLC machine in few days. To avoid confusion, B has been deleted.

Fig. 5: Positions of panels (b) and (c) may be strange. They may be put upper or upper-left from the figures, just like the case of Fig. 2. And, Figure legends for B and C are missing.

Authors’ response:

Thanks for your kind recommendation. The picture has been revised as your suggestion.

Data regarding the enzyme stability in Figs. 5C and 6B: Please show data about “relative enzyme activity” compared without incubation for 12 h before the reaction. In other words, please compare enzyme activity with/without 12-h incubation.   

Authors’ response:

Thanks for your kind recommendation. The picture has been revised as your suggestion. The enzyme activity without 12-h incubation. This was added in Section 2.4 and 2.5.

Table 1: “Optimal pH/temparature” and “pH-stable range” should be defined (criteria for them).

Authors’ response:

Thanks for your kind recommendation. This has been revised under your suggestion.

Lines 269-270: Did the authors just add (but not adjust) each salt into GPPB medium? Because GPPB medium must contain some amount of salts, I suppose this should be described like “xx mM NaCl was “added” Or, did the authors measure/confirm the final concentration of each salt? Please clarify this.

Authors’ response:

Thanks for your kind recommendation. The salts were just added into the GPPB medium without adjustment. Under your suggestion, the description has corrected.

Reviewer 2 Report

An informative manuscript describing characterization of a yeast tannase targeting for biotechnological applications.

It wasn't clear until the methods at the end that the authors did not characterize the native tannase protein as suggested by mention of gel filtration and ion exchange purfication, rather they went the fast route via cloning and yeast expression.  Abstract should be corrected to indicate that efficient approach.  Some information on tanRd gene structure would have be useful.

Also for the SDS-PAGE analysis, this was the recombinant protein?  Did authors try a PAGE zymograph approach to detect the native tannase protein to characterize?

Many rephrasings and typos listed on the attached pdf that will improve readibility of the ms. 

Overall a useful contribution. 

Author Response

An informative manuscript describing characterization of a yeast tannase targeting for biotechnological applications.

Authors’ response:

Thanks for your kind recognition.

It wasn't clear until the methods at the end that the authors did not characterize the native tannase protein as suggested by mention of gel filtration and ion exchange purfication, rather they went the fast route via cloning and yeast expression.  Abstract should be corrected to indicate that efficient approach.  Some information on tanRd gene structure would have be useful.

Authors’ response:

Thanks for your kind recommendation. The abstract was corrected. The sequence of tanRd gene was added.

Also for the SDS-PAGE analysis, this was the recombinant protein?  Did authors try a PAGE zymograph approach to detect the native tannase protein to characterize?

Authors’ response:

   The SDS-PAGE was for the recombinant protein. We did not try a PAGE zymograph approach. As we gained higher activity through the fast route via cloning and yeast expression, potential tannase production can be carried out using the recombinant strain. The characteristics of recombinant tannase should be the concern.

Many rephrasings and typos listed on the attached pdf that will improve readibility of the ms. Overall a useful contribution.

Authors’ response:

Thanks for your kind recommendation. The rephrasings and typos have been corrected under your suggestions.

Have you established it was secreted? And what features of signal peptide indicate a secreted or at least cell surface targeting.

Authors’ response:

   According to the tannase activity determination, the activity was only found in the liquid supernatant of strain Q95, thus this tannase a secreted enzyme. Conducted with the SignalP 4.1 server (http://www.cbs.dtu.dk/services/SignalP-4.1/), the  cutoff site between the signal peptide and mature protein has a rapid change of hydrophobicity.

I didn't see any comparison of native purified TanRd vs the codon optimized cTanRD, or was this done just to get expression ?

Authors’ response:

  The amino acid sequences have not been changed. The codons was optimized just to get expression.

Reviewer 3 Report

The authors describe the isolation and characterization of a novel Rhodosporidium diobovatum Q95 tannase, engineered for heterologous expression in Yarrowia lipolytica. Although the work is a straightforward, the properties reported are routine, and manuscript has some problems and technical issues which need to be improved. The authors should address the following comments and suggestions:
1. The authors have failed to notice the published tannases from Arxula    adeninivorans (Atan1)(B?er et al., Yeast, 26, 323-337 (2009)) and Aspergillus oryzae (TanA and TanB)(Hatamoto et al, Gene, 175, 215-221 (1996) and Koseki et al., J Biosc Bioeng, 126, 553-558 (2018)). These genes have been cloned and characterized. The authors should describe sequence similarity among TanRd and yeast and fungal characterized tannases. Yeast and fungal characterized tannases do not include in Figure 4.
2. Line 114-115; mature protein? Correct molecular weight to molecular mass.
3. Lines 117 and 236; References of tannase and feruloyl esterase family and CAZy database are given in the text.
4. Line 132-133; N-glycosylation recognition sites of TanRd are predicted using NetNGlyc 1.0 server etc.
5. Figure 5; Lane 1, purified Table3 ? Explanation of lane 2 is missing.
6. Line 133; It has been reported that the N-glycosylation has no effect on the activity and stability of AoTanB from A. oryzae (Ichikawa et al., J Biosc Bioeng, 129, 150-154 (2020)).
7. Figure 7; Chemical name of ME
8. Table 2; Kinetic constants toward each substrates should be investigated.
9. Line 224; Correct rhodamine to rhodanine.
10. Some references of journal do not correspond to Marine Drugs style.
11. Line 348; [26]・・・・・・・& Fushinobu, S.

Author Response

The authors describe the isolation and characterization of a novel Rhodosporidium diobovatum Q95 tannase, engineered for heterologous expression in Yarrowia lipolytica. Although the work is a straightforward, the properties reported are routine, and manuscript has some problems and technical issues which need to be improved. The authors should address the following comments and suggestions:

Authors’ response:

Thanks for your kind recognition and recommendation.

  1. The authors have failed to notice the published tannases from Arxula adeninivorans (Atan1)(B?er et al., Yeast, 26, 323-337 (2009)) and Aspergillus oryzae (TanA and TanB)(Hatamoto et al, Gene, 175, 215-221 (1996) and Koseki et al., J Biosc Bioeng, 126, 553-558 (2018)). These genes have been cloned and characterized. The authors should describe sequence similarity among TanRd and yeast and fungal characterized tannases. Yeast and fungal characterized tannases do not include in Figure 4.

Authors’ response:

Thanks for your kind recommendation. The tannases characterized in the articles above have been compared with the tannase in this study. And Yeast and fungal characterized tannases have been enriched in Figure 4.

  1. Line 114-115; mature protein? Correct molecular weight to molecular mass.

Authors’ response: Thanks for your kind recommendation. It has been corrected.

  1. Lines 117 and 236; References of tannase and feruloyl esterase family and CAZy database are given in the text.

Authors’ response: Thanks for your kind recommendation. It has been corrected.

  1. Line 132-133; N-glycosylation recognition sites of TanRd are predicted using NetNGlyc 1.0 server etc.

Authors’ response: Thanks for your kind recommendation. It has been corrected.

  1. Figure 5; Lane 1, purified Table3 ? Explanation of lane 2 is missing.

Authors’ response: Thanks for your kind recommendation. It has been corrected.

  1. Line 133; It has been reported that the N-glycosylation has no effect on the activity and stability of AoTanB from A. oryzae (Ichikawa et al., J Biosc Bioeng, 129, 150-154 (2020)).

Authors’ response: Thanks for your kind recommendation. The discussion has been rewritten to give a better viewpoint.

  1. Figure 7; Chemical name of ME

Authors’ response: Thanks for your kind recommendation. It has been corrected.

  1. Table 2; Kinetic constants toward each substrates should be investigated.

Authors’ response: Thanks for your kind recommendation. Km toward each substrates have been added.

  1. Line 224; Correct rhodamine to rhodanine.

Authors’ response: Thanks for your kind recommendation. It has been corrected.

  1. Some references of journal do not correspond to Marine Drugs style.
  2. Line 348; [26]・・・・・・・& Fushinobu, S.

Authors’ response: Thanks for your kind recommendation.These has been corrected.